# Recent Advances in Improving the Bioavailability of Hydrophobic/Lipophilic Drugs and Their Delivery via Self-Emulsifying Formulations

Rakesh Kumar Ameta [1], Kunjal Soni [1] and Ajaya Bhattarai [2,3,*]

1. Sri M. M. Patel Institute of Sciences & Research, Kadi Sarva Vishwavidhyalaya, Gandhinagar 382023, India
2. Department of Chemistry, Mahendra Morang Adarsh Multiple Campus, Tribhuvan University, Biratnagar 56613, Nepal
3. Department of Chemistry, Indian Institute of Technology, Madras 600036, India
* Correspondence: ajaya.bhattarai@mmamc.tu.edu.np

**Abstract:** Formulations based on emulsions for enhancing hydrophobic and lipophilic drug delivery and its bioavailability have attracted a lot of interest. As potential therapeutic agents, they are integrated with inert oils, emulsions, surfactant solubility, liposomes, etc.; drug delivering systems that use emulsion formations have emerged as a unique and commercially achievable accession to override the issue of less oral bioavailability in connection with hydrophobic and lipophilic drugs. As an ideal isotropic oil mixture of surfactants and co-solvents, it self-emulsifies and forms fine oil in water emulsions when acquainted with aqueous material. As droplets rapidly pass through the stomach, fine oil promotes the vast spread of the drug all over the GI (gastrointestinal tract) and conquers the slow disintegration commonly seen in solid drug forms. The current status of advancement in technologies for drug carrying has promulgated the expansion of innovative drug carriers for the controlled release of self-emulsifying pellets, tablets, capsules, microspheres, etc., which got a boost for drug delivery usage with self-emulsification. The present review article includes various kinds of formulations based on the size of particles and excipients utilized in emulsion formation for drug delivery mechanisms and the increase in the bioavailability of lipophilic/hydrophobic drugs in the present time.

**Keywords:** lipophilic drugs; hydrophobic drugs; emulsion



## 1. Introduction

The improvements in combinatorial chemistry show an enormous rise in a variety of less water-soluble drugs. At least forty novel pharmacologically active lipophilic/hydrophobic moieties exhibit very low aqueous solubility. Nevertheless, a unique challenge regarding drugs is presented to pharmaceutical scientists: orally administrated drugs have innate low aqueous solubility, leading to inadequate oral bioavailability with higher inter- and intra-subject changeability and scarcity of dose proportionality [1]. Numerous formulation perspectives are currently being applied to handle challenges related to formulations of biopharmaceutical class system (BCS) drugs; this includes compound pre-dissolution in suitable solvents followed by capsule filing with this formulation [2], or as solid solution formulations that utilize water-soluble polymers [3]. Although these perspectives will help resolve the matter related to the primary dissolution of drug matter in a liquid phase inside the GI tract up to specific proportions, momentous restrictions such as the precipitation issue of drug molecules in the dispersal of formulations during the crystallization of drugs in the polymer-based matrix are still unsolved. For the same reasons, assessment of physical stability is critical and has been evaluated using techniques such as X-ray crystallography or differential scanning calorimetry. Several formulation modes, such as carrier technology, provide an innovative methodology for enhancing solubility in drug molecules with low

solubilities. Advancements in oral drug molecule bioavailability use lipid-based formulations that have now become attraction points. Perhaps the most adaptable excipient class members presently available are lipids, providing a strong eventuality as a formulator in improving and controlling the lipophilic drug's absorption where ordinary formulation methods failed or when the drug molecule itself is an oil molecule (i.e., Dronabinol, ethyl icosapentate). In addition, with a low affinity for precipitation of lipophilic drugs in the GI tract during dilution, such formulations will favor partitioning kinetics in the lipid droplets to be retained [4]. The literature review indicates that the application of carrier technology is a perspective of scientific interest in lipid-based oral formulations and strengthens the ambidexterity in addressing the issues complementary to oral drug delivery of poorly soluble molecules [2–6]. Novel methods such as self-emulsification modes have also intensified the solubility of inadequately soluble drugs and have some advantages. The introduction of this self-emulsification concept and present-day advances in polymer science have led to application advancements with lipid-based self-emulsifying formulations in various drug delivery views comprising drug targeting. This article endeavors to review the far-reaching awareness of emulsion-forming drug delivery systems (DDS) for the bioavailability enhancement of hydrophobic/lipophilic drugs by cherishing numerous formulations. The present-day architectural innovations and the advancement of self-nano-emulsifying and self-micro-emulsifying formulations have also been considered. Thus, this review's main focus is on improving the bioavailability or solubility of hydrophobic/lipophilic drugs via self-emulsifying formulations (SEF).

## 2. Emulsion Concept and Types of Emulsions

An isotropic and transparent solution from an oil mixture and co-solvent, with surfactant and co-surfactant, is emulsified with gentle agitation equivalent to that experienced in the GI tract; this is known as SEF. Spontaneous emulsification is recognized for this solution in aqueous GI fluids in the presence of oral administration. Bile secretion is stimulated by this triglyceride (emulsified oil), which further emulsifies oil droplets containing the drug. Lipases and co-lipases, secreted from various portions such as the salivary gland, pancreas, and gastric mucosa, then metabolize these lipid droplets and hydrolyze the triglycerides by forming free fatty acids and di- and mono-glycerides. Additionally, these molecules get solubilized when they pass through the GI tract. In due course, emulsion droplets are formed in various sizes, along with mixed micelles and vesicular structures containing phospholipids, bile salts, and cholesterol [5]. The synthesis of chylomicron occurs in lymphatics, ensuring enhanced drug absorption. The bioavailability intensifies formulations' self-emulsification characteristics primarily in connection with confident in vivo features, such as inhibiting cellular efflux mechanisms and retaining the drugs from circulation; this is because of the attachment of several lipidic excipients with a particular drug, a decrease in drug metabolism by the liver in the first pass, as well as its uptake in the lymphatic transport system. In addition, the formation of micellar suspensions and fine dispersions that restrict recrystallization and/or precipitation of drug molecules, where changes in the GI fluid begin because of the properties of several lipid components that favor upgraded drug absorption [6]. Usually, emulsion-forming drug delivery systems (EFDDS) are prepared as simple emulsions, whereas surfactants with a hydrophilic–lipophilic balance (HLB) < 12 SEFs are formulated. Self-nano-emulsifying formulations (SNEFs) and self-micro-emulsifying formulations (SMEs) are acquired using surfactants of HLB > 12. Because of surface enhancement for dispersion, the formulations contain improved dissolution (drugs with poor solubility) and high stability. Due to this, independent drug absorption from bile secretion ensures a speedy shift over that of less soluble drugs in blood. Additionally, their formulations have specific, definite characteristics associated with upgraded drug delivery systems. Thus, the emulsion contains hydrophobic and hydrophilic parts.

### 3. Excipients for Self-Emulsifying Formulations

Based on studies, the process of self-emulsification is precise to the kind of surfactant/oil pair utilized, the concentration of surfactant, the ratio of oil/surfactant, and the temperature at which the self-emulsification materialized. These salient findings have been supported by the fact that only selective combinations of pharmaceutical excipients led to effective systems of self-emulsifying therapeutics. Numerous remarkable components are used in EFDDS, such as the following.

### 3.1. Oils

The most natural support for lipid vehicles is available from consistent edible oils. However, they have poor dissolving properties for enormous amounts of hydrophobic drugs, and limitations for superior self-micro emulsification considerably decrease their use in SEFs. Vegetable oils that are altered or hydrolyzed have a widespread role in successive SEFs attributed to their biocompatibility. Naturally occurring di- and triglycerides have been used exponentially as excipients with susceptibility for degradation (a crucial pathway for releasing drug molecules from EFDDS-based formulations). At present, triglycerides with a medium chain are being replaced by new semi-synthetic medium-chain-containing triglycerides, with molecules such as Gelucire. Robust emulsification systems are formed based on their ability for notable fluidity and the remarkable solubilizing prospect capabilities of self-emulsification. Oils and fats, such as corn, olive, palm, soya bean oils, and animal fats, either digestible or non-digestible, may be used as oil phases in EFDDS [7].

### 3.2. Surfactants or Emulsifiers

Based on the critical packing parameter and hydrophilic–lipophilic balance, the screening of surfactants can be carried out. For the formation of EFDDS, non-ionic surfactants are often chosen because of their limited toxicity; furthermore, they have reduced critical micelle concentrations compared with their ionic complements [8]. High HLB-value-containing surfactants are commonly used in forming EFDDS, including polysorbate 80, poloxamers, Gelucire (HLB 10), sorbitan monooleate (Span 80), cremophor EL, hexadecyltrimethylammonium bromide, sodium lauryl sulphate, and bis2-Ethylhexyl sulfosuccinate. Additionally, fatty alcohols and famous surfactants, such as cetyl and stearyl, lauryl, glyceryl, and esters of fatty acids, are also incorporated [9]. Surfactants occurring naturally are also endorsed for SEFs; the most commonly used surfactant is lecithin, with the due reason of the most significant biocompatibility. It has phosphatidylcholine as a fundamental component, having an amphiphilic structure and water-solubilizing properties. To get stable SEFs, the commonly used surfactant concentration is 30 to 60% $w/w$.

A higher surfactant concentration (~60%) may cause selective, reversible alterations in intestinal wall permeability or GI tract irritability. The subsequent hydrophilicity and higher HLB of surfactants are essential for the prompt formulation of $o/w$ droplets and/or for the sudden spread of the formulate in an aqueous condition, yielding exceptional self-emulsifying/dispersing achievement. By nature, surface-active agents are amphiphilic and generally dissolve more remarkably than hydrophobic drugs. This property is of great importance as it prevents precipitation through the GI lumen, and drug molecules have prolonged presence in a solubilized form, an essential phase for effective absorption [10]. Recent reports show that the surfactant digestion mechanism affects the performance of SEFs, as the dissolving environment alterations can cause precipitation of a little less water-soluble drugs.

Additionally, more information is needed for degradation molecule formation from surfactants and their interaction with fatty acids and phospholipids, bile salts, and dietary lipids such as endogenous lipids. This may play a vital role in maintaining a solution with poorly water-soluble drugs, and an essential part in forming mixed micelles may be compromised [11–13]. Considering all these data reported, information about the impacts of non-ionic surfactants that may show inhibition of triglyceride digestion is vital for lipid-based formulation devel-

opment. In addition, the susceptibility of surfactants towards digestion through pancreatic enzymes is an essential factor that should be considered in formulation development.

A simple mechanism/preparation of a self-emulsifying system is depicted in Figure 1, which could help the reader understand this.

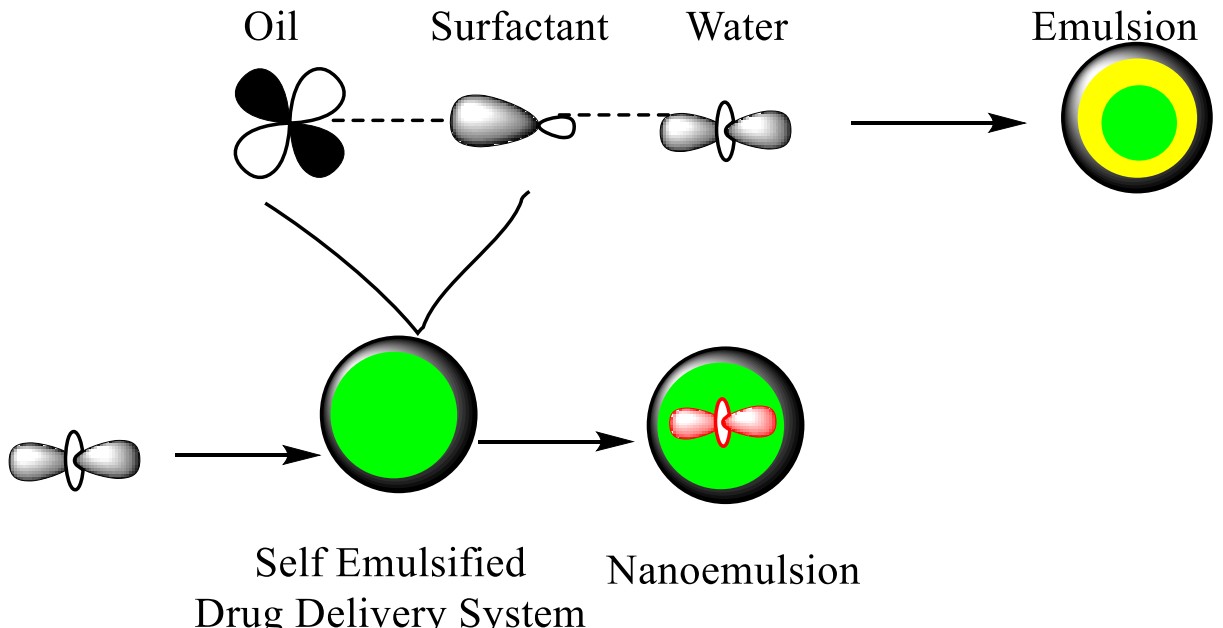

**Figure 1.** Pictorial representation of a self-emulsified drug delivery system and nanoemulsion.

### 3.3. Co-Surfactants/Co-Solvents

Frequently, a cosurfactant is added in the self-emulsifying formulations to increase interfacial area and dispersion entropy and decrease free energy at its minimum and interfacial tension [8]; on the cause of amphiphilic nature, this is substantially accumulated by a cosurfactant at an interfacial layer that increases interfacial film fluidity through surfactant monolayer penetration. Pentanol, hexanol, and octanol, like short and medium-chain alcohols, are preferred cosurfactants known to form self-emulsifying formulations spontaneously. Apart from cosurfactants, transcutol (diethylene glycol mono ethylene ether), triacetin (an acetylated derivative of glycerol), polyethene glycol, propylene glycol, propylene carbonate, glycofurol (tetrahydro furfuryl alcohol polyethene glycol ether), etc., like several co-solvents, are convenient for hydrophobic drug dissolution in lipid bases. Recently, with varying fatty acids, polyglycolyzed glycerides (PGG) and polyethene glycol (PEG) in aggregation with vegetable oils have also been reported for use in hydrophobic drug solubilization and emulsification [7,14,15].

Thus, the structure and stability of *w/o* and *o/w* types of emulsions are mainly based on the constituents of the emulsifier and how much hydrophobicity or hydrophilicity it has.

### 4. Recent Studies

### 4.1. Self-Stabilized Pickering Emulsion

A novel high-pressure homogenization technique for silybin oral bioavailability enhancement has been developed for silybin nanocrystal self-stabilized Pickering emulsion (SN-SSPE). The impacts of drug content and homogenization pressure on SN-SSPE formation were also evaluated. Using SEM (scanning electron micrograph), atomic force microscopy (AFM), and confocal laser scanning microscopy, the structure, size, and morphology of PE droplets were identified. Investigation of in vivo oral bioavailability and SN-SSPE release in vitro was also carried out. The results revealed that when homogenization pressure is scaled up to 100 MPa, the silybin nanocrystals' (SN-NC) particle size decreases. A stable silybin Pickering emulsion might be formed when silybin content

reaches 300 mg or above; thus, surfaces of oil droplets are entirely covered by sufficient SN-NC, and a self-stabilized Pickering emulsion is formed. A core-shell arrangement composed of a core of SN-NC shell and oil was seen when the SN-SSPE emulsion droplet was $27.3 \pm 3.1$ μm. A stability of about 40 days or more has been recorded for SN-SSPE. SN-SSPE showed a faster in vitro release rate compared with silybin coarse powder. However, it is similar to the suspension of SN-NCS. Intragastric administration showed a 2.5-fold and 3.6-fold incremental peak concentration of SN-SSPE of silybin compared with SN-NCS and coarse powder of silybin in rats. Furthermore, there were 1.6-fold and 4.0-fold increases in the AUC of SN-SSPE concerning SN-NCS and silybin coarse powder. From the results, it has been confirmed that silybin nanocrystals could stabilize the Pickering emulsion of silybin and increase oral bioavailability. The self-stabilized Pickering emulsion of drug nanocrystals has encouraged a system for poorly soluble drugs for oral drug delivery [16]. For a controlled drug delivery system, (PLGA) (Poly lactide-co-glycolide) microparticles are often utilized. To formulate PLGA microparticles, standard emulsion methods have been used. However, they show less loading capacity, more precisely for poorly soluble drugs in organic solvents. A template of water-soluble polymers and nanocrystal technology was used to manufacture nanocrystal-loaded microparticles with enhanced encapsulation and drug-loading efficacy for extended breviscapine delivery, as reported by Hong Wang et al. [17]. A precipitation-ultrasonication method is used to prepare breviscapine nanocrystals, which are cast using water-soluble polymer mould load further into PLGA microparticles. These disc-like particles were characterized and compared with spherical particles by emulsion-solvent evaporation. Through confocal laser scanning microscopy (CLSM) and X-ray powder diffraction (XRPD), analysis of the highly dispersed breviscapine state in microparticle confirmation is carried out. The drug significantly affects the efficiency of breviscapine and its loading capacity in PLGA microparticles and liberation mechanism through loading percentage and fabrication methodology. When both template and nanocrystal methods were enforced, drug loading and encapsulation potential was enhanced by 2.4% to 15.3%, and 48.5% to 91.9%, respectively.

### 4.2. Drug Loaded Nanoemulsions

On the other hand, as drug loading is increased, loading efficiency is reduced. An initial release by bursting in all microparticles has been seen that later slows down to 28 days and is further followed by erosion-acceleration phase release, which supports sustained delivery over a month for breviscapine. Stable serum drug level after intramuscular microparticle injection in rats was observed even after 30 days. Therefore, nanocrystals of less-soluble drug-loaded PLGA microparticles provide a supportive way for long-term therapeutic outcome characterization desirable in vivo and in vitro accomplishment. An approved safe herbal drug for several hepatic disorders is Silymarin. However, poor oral bioavailability is its major limitation. Silymarin-loaded nanoemulsions could be prepared using the high-pressure homogenization (HPH) method [18]. Capryol 90 as the oil phase, Solutol HS 15 as a surfactant, and Transcutol HP as a co-surfactant were selected accordingly. Design-based quality has been employed for optimized nanoemulsions in conditions of several cycles, processing pressure, and (Smix) surfactant/co-surfactant mixture amount. Globule size, polydispersity index (PDI), zeta potential, transmittance, and percentage in vitro drug release of the optimized formulation were found as $50.02 \pm 4.5$ nm, $0.45 \pm 0.02$, $-31.49$ mV, $100.00 \pm 2.21\%$ and $90.00 \pm 1.83\%$, respectively. The apparent permeability coefficient (Papp) has been enhanced by nanoemulsion, as shown in everted gut sac studies. Silymarin Papp in nanoemulsion and the oral suspension was $1.00 \times 10^{-5}$ cm/h with a flux of $0.422$ μg/cm$^2$/h, and $6.30 \times 10^{-6}$ cm/h with a flux of $0.254$ μg/cm$^2$/h at 2 h, respectively. The enhancement in Silymarin bioavailability in nanoemulsion was compared with its oral suspension. Silymarin nanoemulsion could be an excellent oral delivery system with improved oral bioavailability that was found to be significant ($p < 0.05$) in a pharmacokinetic study.

As a unique lipid excipient class, phosphorylated tocopherols have exhibited potential in pharmaceutical utilization. They can make poorly water-soluble drugs soluble, which demonstrates a potential advantage in enhancing drug bioavailability where solubility is a limiting factor. A formulation of CoQ10, combined from medium-chain triglyceride (MCT) and phosphorylated tocopherols, TPM, and their in vivo and in vitro functionality was compared with tocopherol-based additional alternative excipients studied. CoQ10 was less soluble in digesting MCT during in vitro digestion experimentation as an anticipated molecule. TPM addition facilitated improved CoQ10 solubilization as vitamin E TPGS did. Several other derivatives of tocopherol, such as tocopherol and tocopherol acetate, were found to be less impactful at active solubilizing during digestion. In vitro solubility trends were preserved during in vivo CoQ10 bioavailability after oral administration in rats, where TPGS and TPM formulations provide nearly double exposure of MCT apart. Simultaneously, the overall exposure is reduced by the addition of other tocopherol derivatives. The resultant condition reveals TPM as a potent new solubilizing excipient for drugs with low water-solubility in oral drug delivery [19]. Δ9-tetrahydrocannabinol (THC) and cannabidiol (CBD), a lipophilic phytocannabinoid, represent therapeutic potential in numerous medical situations. Both compounds have low water solubility and are subjected to comprehensive first-pass metabolism in the GI, leading to limited oral bioavailability of hardly up to 9%. An advanced drug delivery system with lipid-based self-emulsification has been developed by Irina Cherniakov et al. and termed an advanced pre-concentrate of Pro-NanoLiposphere (PNL). Lipids and emulsifying excipients of GRAS compose PNL and are known for solubility enhancement with decreased phase I metabolism of lipophilically active compounds. When panels are incorporated with a natural absorption improver, they become advanced PNLs. They are natural phenolic compounds and alkaloids reported as inhibiting phase I and II metabolism processes. Hence the use of these advanced-PNLs on THC and CBD oral bioavailability has been explored. A 6-fold increase is reported in AUC compared with CBD solution when CBD-piperine-PNL has been orally administered, indicated as the most potent screened formulation. Similar data was found during THC-piperine-PNL-based pharmacokinetic experiments, showing a 9.3-fold increment in AUC compared with the THC solution. The synthesized Piperine-PNL can synchronize piperine with THC or CBD delivery to the enterocyte site. The bioavailability of CBD and THC has increased due to this co-localization and the effect on pre-enterocyte and enterocyte levels during the absorption process. The further amplification in THC and CBD absorption is incorporated by piperine into PNL. It plays a role in phase I and phase II metabolism inhibition by piperine with an addition to P-gp and phase I metabolism by PNL. These offbeat results put forward the way for piperine-PNL to deliver less soluble, highly metabolized drugs that cannot be orally administered at present [20].

### 4.3. Nano Lipospheres Formulation

In pomegranate, ellagic acid is a predominantly bioactive compound with low bioavailability. A food-grade system from self-nano emulsification has developed that enhanced the dissolving and absorption of ellagic acid. Pseudo-turning phase images and solubility assays have revealed that the components are suitable for formulation. The optimal formulation has been achieved with polyethylene glycol, polysorbate, and capric triacylglycerol/caprylic at 45/45/10 wt.%. A fine nanoemulsion was yielded from controlled stirring and optimized formulation, with an average droplet size of 120 nm. With the formulation, the ellagic acid dissolution was remarkably increased. Based on the pharmacokinetics study carried out in rats, ellagic acid's bioavailability was 3.2 and 6.6-fold higher compared with its aqueous suspensions and pomegranate extract. A novel strategy has been developed to deliver ellagic acid with a self-nano-emulsifying method for developing dietary supplement products and ellagic acid-containing functional foods [21]. The most rapidly growing therapeutic segment is biologics, but it has limitations of low stability. It has an alternative delivery system for personal administration; however, it has particular physiological challenges that prompt protein susceptibility and function loss. Protein formulation

in biomaterials, such as electrospun fiber, can resize these barriers. Still, optimization of such platforms is required for protein stability and maintenance of bioactivity during the formulation process. An emulsion electrospinning method has been developed for protein loading into Eudragit L100 fibers for perioral delivery. Alkaline phosphatase and horse reddish peroxidase encapsulation lead to higher efficiency into fibers and pH-specific release. Protein bioavailability recovery has been enhanced by aqueous emulsion phase reduction and hydrophilic polymer excipient inclusion. Hannah Frizzell et al. demonstrated that protein formulation in lyophilized electrospun fibers increases therapeutic compounds' shelf life compared with aquatic storage. Thus, a novel promising dosage form of biotherapeutics for perioral delivery has been available from the platform [22]. In another work, designing a new octa-arginine (R8) altered with (LE) a lipid emulsion system of the lipophilic drug disulfiram (DSF) for ocular delivery was the target purpose. On corneal permeation, R8 presence and lipid emulsion particle sizing (DSF-LE1, DSF-LE2, DSF-LE3) with DSF loading and altered with R8 (DSF-LE1-R8 and DSF-LE2-R8) was formulated. There was a change in zeta potential from negative to positive values for lipid emulsions after the modification of R8.

DSF-LE1-R8 fabricates the strongest mucoadhesion from different compositions of mucoadhesion studied. R8 altered lipid emulsion (DSF-LE1-R8) with nano-sized particles showed ocular distribution in vivo and corneal penetration in vitro, with high permeability, and the most significant DDC amount distribution in visual tissues. More homogeneous fluorescence was displayed by LE1-R8 when LE1-R8 was labelled with Coumarin-6 and deep penetration in the cornea compared with other formulations at different time frames. Furthermore, LE-R8 could be transported across the corneal epithelium apart from its paracellular routes by using transcellular ways because of an induced update on a cause of R8 modification and confirmation received by using confocal laser scanning microscopy. It was also reported that DSF-LE1-R8 exhibits a marked anti-cataract effect from evaluation. Hence, nano-sized particles with R8 alterations in lipid emulsions were proposed as a significant ocular delivery method to enhance penetration in the cornea and DSF visual delivery [23]. Capsule designing with insulin-like hydrophilic drugs for the preservation of its biological activity and stability through double emulsion methodology and its entrapment into biodegradable microcapsules in the following manner with xanthan and chitosan gum complexes containing shells was devotedly investigated by Mutaliyeva et al. Several formation factors such as biopolymer and oil type, stabilizer, and its concentration, aqueous phase internal content, volume fraction, regime mixing, and time were also evaluated. The complex's effects on the emulsion formation process, characteristics, and stability of resultant emulsions were interrogated using interfacial charge (zeta-potential) and the size distribution (DLS). The prepared capsules were analyzed using size distribution, zeta potential, and microscopic characterizations. Insulin release kinetics was monitored through UV–vis spectroscopy, and reports suggested that sustainability enhancement was progressive [24].

### 4.4. W/O/W Double Emulsion Formulation

Sesamol, the phenolic compound and degradative product of sesamolin, has poor bioavailability but is recognized for its anti-inflammatory properties. An attempt was made to increase its bioavailability through mixed phosphatidylcholine micelles encapsulation. Sesamol solubilization and entrapment can be seen in PCS (phosphatidylcholine mixed micelles), having a 3.0 nm particle size with 96% efficiency. PCS showed lower comparative fluorescence intensity when it was compared with free sesamol. PCS cellular uptake, bioaccessibility, and transport across cell monolayer was 1.2-fold, 8.58%, and 1.5 times improved compared with FS. By using lipoxygenase inhibition and an LPS-treated RAW 264.7 cell line, the FS and impact of PCS regarding its anti-inflammatory action were studied. The iNOS protein expression downregulation (27%), ROS (32%), NO (20%), and inhibition of lipoxygenase with 31.24 μM ($IC_{50}$) were affected by PCS in comparison to FS [25]. Omid Shamsara et al. loaded piroxicam (PX) into multi-layered oil-water emulsions and

stabilized using the complexes of β-lactoglobulin (β-L) and pectin, in which homogenized sunflower oil was used as a primary emulsion with a β-L solution containing PX. These droplets get stabilized by a subordinate layer of pectin. The low-methoxyl sunflower pectin (LMSP), low-methoxyl citrus pectin (LMCP), high-methoxyl apple pectin (HMAP), and high-methoxyl citrus pectin (HMCP), were respectively utilized to produce emulsion droplets with a secondary or double layer. Creaming stability, PX entrapping efficiency (%), and droplet mean size of emulsion (D43) have been determined. Trend or PX release was examined and used for the zero-order kinetic experiment. The output of such experimental work has suggested that citrus pectins with NaCl and β-L/high-methoxyl-apple-stabilized emulsions were found with maximum stability, with good PX loading capacity compared with stabilized emulsions by a β-L complex of pectin in the absence of NaCl. The mean droplet size of double-layered emulsions increased at a high pectin fraction and reduced by a low β-L fraction [26].

A potent bioactive molecule such as betulinic acid (BA) is recognized for therapeutic action. Yet, it has limited efficacy because of poor solubility and low bioavailability. Harwansh and coworkers developed BA-loaded nanoemulsions with increased hepatoprotective activity and bioavailability. Using the BA-NE1 procedure, the nanoemulsion was formulated containing surfactants such as labrasol, olive oil, aqueous phase, and co-surfactant, such as plural isostearate, in the convenient ratio optimized. Its characterization was done through several parameters, such as the size of the droplet, refractive index, zeta potential, FTIR, UV-spectrophotometry, TEM, and stability studies. A droplet size of about 150.3 nm with negative zeta potential such as $-10.2$ mV of this emulsion was evaluated. Pharmacokinetic limits such as Cmax (96.29 $ngmL^{-1}$), AUC0-t ∞ (2540.35 $nghmL^{-1}$), the elimination half-life (11.35 h), Tmax (12.32 h), and relative bioavailability (440.48%F) were also investigated and compared with BA. The hepatic serum marker levels and antioxidant enzymes concerning CCl4-intoxicated groups (** $p < 0.05$ and *** $p < 0.01$) were significantly restored by BA-NE1. Accordingly, the study also reveals that the BA-loaded nanoemulsion could improve hepatoprotective activity because of increased solubilization and enhanced oral bioavailability [27]. There is an advanced formulation with better biocompatibility, stability, and higher loading of hydrophobic drugs in submicron emulsions (SEs) from sterilization with an autoclave. To increase the targeting and uptake of tumor cells, SEs get altered by target moieties and a positive charge. Cationic DocSEs (DocCSEs), docetaxel-loaded SEs (DocSEs), and peptide-RLT-modified DocCSEs targeted by low-density lipoprotein receptor (LDLR) were formulated. A particle size of $182.2 \pm 10$ nm and loading efficiency of docetaxel (Doc) 98% with a zeta potential of $39.62 \pm 2.41$ mV was reported for optimized RLT-DocCSEs. They showed 96 h of sustained release and were found stable for 2 months at 4 degrees Celsius. Significantly more cell apoptosis and RLT-DocCSEs have caused inhibition of cells compared with DocCSEs and DocSEs. RLT-DocCSEs showed greater cellular uptake with slow elimination from DocCSEs and DocSEs [28].

### 4.5. Polymeric Emulsifier Containing Formulation

Polyethylene glycol (PEG), lactide (LA), and ε-caprolactone (CL) were derived as amphiphilic bioresorbable copolymers studied for their emulsification and degradation properties. With monomethoxy PEG, lipophilic 20 wt.% PCL, PLACL block, PLA, and 80 wt.% PEG (hydrophilic) block comprising polymers were formulated with the LA and/or CL ring-opening polymerization process. These emulsifiers have analogous capabilities for stabilizing squalane/water interfaces in emulsification as they possess equivalent hydrophilic–lipophilic balance (HLB) values. Polymer degradation within the emulsion and in the aqueous phase at 37 degrees Celsius to mimic conditions of the human body was carried out. According to the result, the polymer degradability was found to cause instability in the emulsion. In addition, emulsion polymer matrices exhibit lower degradative rates than in an aqueous phase from corresponding polymers. The characteristics in pharmaceutical applications are of keen interest, particularly for sustained delivery mechanism designing [29]. A study aimed to develop a re-dispersible dry emulsion that

contains simvastatin, a model drug with lipophilic, low water solubility properties; they used a fluid bed coating methodology. This represented manufacturing of dry-emulsion-mode-formulated pellets, in which a dry emulsion layer was applied to a neutral core. As an oily phase, 1-oleoyl-rac-glycerol, a preliminary formulated material, was selected because of its higher drug solubility and potent bioavailability possibilities. Tween 20, mannitol, and HPMC were used as solid surfactants and carriers. The experimental design was used more specifically for mixture design to get the optimal formulation composition. The initial responses used as formulation optimization parameters were the stability and ability to reconstitute the emulsion. On optimization, the formulation represented slender-sized droplet distribution at reconstitution, high strength, satisfactory drug encapsulation, and an increase in dissolution possessions as compared with a pure drug and a non-lipid-based tablet. Uniform morphological data for the functional layer and separated droplets with simvastatin and uniform distribution of size and coated pellets with a circular shape were derived from image analysis using Raman spectroscopy and scanning electron microscopy. The work revealed the evidential design concept of re-dispersible dry emulsions using the fluid bed layer technique [30].

### 4.6. Nano-Precipitation/Dry Formulations

From another work, nanoemulsion from surfactant-free Pickering formulations can liberate a drug with enhanced oral bioavailability at specific pH. By using the nano-precipitation method, magnesium hydroxide-based stabilizing nanoparticles were obtained. The $Mg(OH)_2$ nanoparticles stabilized oil-in-water Pickering nanoemulsions were prepared using a high-energy procedure and sonication probe. The effect of all formulating properties, composition, and the $Mg(OH)_2$ nanoparticles' size on the physicochemical parameters of Pickering nanoemulsions was explored with experimental processes. By using transmission electron microscopy and DLS, the formation was characterized. Moreover, the $Mg(OH)_2$ was solubilized in an acid medium as an advantage that leads to nanoemulsion destabilization and oral release of active components. It is revealed from the acid-releasing work (pH = 1.2) that an increase in release is due to the loading of nanodroplets with the saturation of concentration. At pH = 6.8 (an alkaline media), ibuprofen is significantly released from saturated nanoemulsions in an acid medium. These nanoemulsions not only prompt drug bioavailability but also protect patients from acid medicine side effects through the basic features of hydroxides. Additionally, hydroxides increase pH when present in the stomach; enhancing the release of ibuprofen is greatly affected by pH for solubility [31].

The drug atorvastatin calcium (ATV) is less bioavailable. A dry emulsion method was orally utilized with lyophilized disintegrating tablet development to improve its dissolution in vitro and performance in vivo. Under proper homogenization, the emulsions were formulated using a collapse protectant (glycine) as an aqueous phase, 4% alginate/gelatin-containing mannitol, and synperonic PE/P 84 (surfactant) as an oil phase. The impact of the emulsion formulation parameters was investigated for the prepared tablets' friability, in vitro dissolution, and disintegration time for tablets to the drug. The outcomes revealed the important impact of matrix emulsifier types and the former on disintegration time. From a study of in vitro dissolution, the ATV rate of dissolution was enhanced from lyophilized-dry-emulsion-tablets (LDET) in comparison with a plain drug. Optimized ATV-loaded LDET was studied for DSC and XRD, and the results proved drug presence in the amorphous form. From the SEM images, the intact, non-collapsible, porous-structured LDET was seen to have a complete ATV crystallinity loss. When high-fatty rats were administrated with ATV-loaded LDET, the serum and tissue levels were found to be significantly decreased [32]. The polymers that have grown extensively in the last decades are known to be smart polymers because of their extensive uses for drug targets with controlled drug delivery methods. Based on this concept, Chekuri Ashok et al. used *Albizia lebbeck* L. seed polysaccharide (ALPS) to design and make the preparation of smart releasing emulsion (*o/w*). Similarly, the physicochemical properties, such as the capacity of emulsion (EC), viscosity, stability of emulsion (ES), polydispersity index (PDI) zeta potential, and related

parameters were examined. The EC/ES was found to increase with the increase in ALPS concentration. Using factorial design possibilities, emulsion formulations were statistically oriented. The shear-thinning behavior was seen in all the emulsions. The polydispersity index and zeta potential were recorded at 0.232–1.000 and −35.83 mV to −19.00 mV, respectively.

### 4.7. Solid-Self-Nano/Micro-Emulsifying-Drug-Delivery-Systems

Moreover, the cumulative percent drug release at 8 h from the emulsions was between 30.19–82.65%. The zero-order release kinetics was observed for the drug release profile. So, as a conclusion, the ALPS could be used as a smart polymer and a natural emulsifier to prepare pH-sensitive emulsions for drug delivery systems [33]. Parth Sharma and coworkers demonstrated that various solid-self-nano-emulsifying-drug-delivery-systems (S-SNEDDS) could be prepared using porous hydrophobic and hydrophilic carriers to enhance the simvastatin (SIM) bioavailability and dissolution rate. When 0.1% SIM is used to prepare SNEDDS containing Labrafil M 1944 CS, Tween-80, and ethanol, the resultant droplet is 40.69 nm. Aerosil-200, Syloid XDP 3150, Micro Crystalline Cellulose PH102, Syloid 244FP, and lactose were applied as hydrophobic carriers, whereas CMC, sodium carboxy methyl cellulose, hydroxyl propyl-β-cyclodextrin, and polyvinyl alcohol were used as hydrophilic carriers. Through biopharmaceutical, micrometric, and stability studies, S-SNEDDS was characterized. The S-SNEDDS and liquid-SNEDDS of Aerosil 200 have shown significant superiority on unprocessed and marketed SIM from in vitro dissolution determinations. The crystalline SIM was revealed through a DSC, XRD, and scanning electron microscope when present in altered amorphous SNEDDS preparations, where Aerosil 200 was used as a carrier. In addition, pharmacokinetic studies carried out on rats demonstrated an increase in time of 0.5 h for (Tmax) maximal concentration, maximal concentration (Cmax) by 3.75 folds, mean residence time with 1.22 h, (AUC0-t) area under the curve by 1.54 times, AUC0-∞ with 2.10 folds, and bioavailability by 3.28 folds. All these findings support the superiority of developed S-SNEDDS over the market formulation. It can be concluded that such S-SNEDDS enhances the SIM's bioavailability and dissolution rate [34].

Another piece of work sought to investigate excipient influences on liquid self-micro-emulsifying-drug-delivery-systems (SMEDDS) and l-tetrahydropalmatine (l-THP) containing SMEDDS laden in the pellet properties. In addition, the study was extended with a rabbit model to compare l-THP suspension and such SMEDDS bioavailability. Capryol 90 and surfactant mis were interrogated in the SMEDDS formulation at their optimum ratio. In pellet-SMEDDS, l-THP showed an amorphous state proved by powder X-ray diffractometry. When a pharmacokinetic study using LCMS spectrometry was carried out in a rabbit model, the results revealed that SMEDDS enhances l-THP oral bioavailability by 198.63% compared with the l-THP suspension. The study also demonstrated that there was no noteworthy difference from the original liquid SMEDDS, the ultimate mean concentration (Cmax), and the relative mean bioavailability of pellet-SMEDDS [35]. Another potent anti-inflammatory agent, *Boswellia serrate* gum resin, has been vastly used in ancient medicines. However, its efficacy must be evaluated, as it has low oral bioavailability. Hence, a self-nano emulsifying system (SNES) has been used to improve the systemic concentration of boswellic acids. The (KBA) 11-keto-β-boswellic acid and (AKBA) acetyl-11-keto-β-boswellic acid are the most biologically active constituents of boswellic acids used for indication and evaluation of the efficiency of the self-emulsifying system. In a lipolysis study, KBA and AKBA bio-accessibility and aqueous solubility were reported to increase by 2.3 and 2.7-fold, respectively. A noteworthy increase in the oral bioavailability of these two acids was recorded by more than two, as compared with bulk oil suspension from an in vivo pharmacokinetic study. Thus, SNES is an effective oral formulation with more storage stability to improve the bioavailability of boswellic acids [36].

### 4.8. Self-Nanoemulsifying System to Improve the Oral Bioavailability

Curcumin and coumarin are famous for their broad spectra pharmacological and biological activities, such as antioxidant, anti-inflammatory, anticancer, and antimicrobial,

with impaired therapeutic administrations due to poor solubility and less stability in water. Based on the encapsulation efficacy (EE; the loaded amount of drug into formulations) of such drugs, their bioavailability is assessed. Hence, work was carried out to overcome these limitations by enhancing bioavailability with nano-encapsulated emulsions. Aminated nano cellulose (ANC) particles can stabilize the PE through an oil/water-based process for complete factorial optimization design from different components of the oil phase with Tween 80 and medium-chain triglyceride (MCT) for nanoemulsions. The PEs and nanoemulsion formulations were obtained with a particle size of $\leq$150 nm. Along with zeta potentials, the storage time, pH, and ANC concentration on emulsion stability as influencing factors were examined. Curcumin and coumarin's EE were found to be > 90%. The kinetic profiles of the encapsulated PEs' release demonstrated a sustained release with possible increased bioavailability. Curcumin-encapsulated PE showed a high release percentage compared with coumarin. Curcumin and coumarin-loaded PEs were also examined for antimicrobial as well as anticancer potential using Gram (+)/($-$) bacteria and fungi and L929 and MCF-7 (human cell lines), respectively, in in vitro cytotoxicity determination. The results proved that PE curcumin and coumarin are significant inhibitors of microbial growth and prevention from cancer [37]. Ibuprofen is a potent analgesic and a non-steroidal anti-inflammatory drug, where the administration can pose side effects or reactions in the body, such as ulcers or bleeding, and can lead to increased stomach or intestinal perforation risk. In a report, Yiping Deng et al. reported IBU nanoparticles (IBU-NPs) formulation by emulsion-solvent-freeze-drying/evaporation to improve its solubility. Under optimum conditions, IBUNPs were produced with a $216.9 \pm 10.7$ nm particle size, which was characterized by DSC, X-ray, SEM techniques, equilibrium solubility, in vitro transdermal rate, and transdermal bioavailability. Morphological features of IBU-NPs revealed porous clusters. From the analysis of prepared IBU-NPs, a low crystallinity was observed. Chloroform and ethanol residual amounts were 9.6 and 170 ppm, respectively, lesser than the class II ICH limit. The IBU-NPs chemical structure was retained, but IBU-NPs, after preparation, underwent amorphous states, reported from measurement analysis. Compared with transdermal and oral raw IBU, IBU transdermal bioavailability was significantly enhanced by the IBU-NP group. Moreover, IBU-NP transdermal gel exhibited a stable cooling rate and longer cooling duration in febrile rats. Even at low and mid doses, better efficacy was given by IBU-NP transdermal gel than oral IBU. The results concluded that for transdermal delivery formulations, IBU-NPs could be appliable and have potent value for non-oral administration [38]. In different kinds of cancer treatment, colloidal particles (CPs) are developing materials in drug transport as they have rapid effectiveness and biosafety. Rose PLGA/Bengal CPs were formulated by W/O/W emulsion and layer-by-layer electrostatic adsorption. Furthermore, they were evaluated and characterized as having potential for breast cancer treatment. They were also examined for the efficacy of zeta potential, drug release kinetics, size, HCC70 (negative breast cancer cell line), and cell viability inhibition. The outcomes revealed that all kinds of CPs may be an alternative to routine cancer treatment as having enhanced retention (EPR)/permeation impacts solid tumors. However, CPs with W/O/W double emulsion showed delivery times of up to 60% in 2 days, which is more suitable, whereas layer-by-layer CPs displayed a t release of 50% in just 90 min. Cell viability could be decreased with both types of CPs, which was encouraging for in vivo testing models that can help prove their feasibility and efficacy against triple-negative breast cancer treatment [39].

With high-pressure homogenization, corn oil and polysorbate 80 tea polyphenols (TP) were emulsified. The $99.42 \pm 1.25$ nm sized droplet for O/W TP nanoemulsion was reported on the formulation. In storage at 4/25/40 °C, TP nanoemulsion was found to be stable. A simulated digestion assay in an in vitro study found that ($-$)-epigallocatechin gallate (EGCG) bio-accessibility was enhanced in a nanoemulsion than in aqueous solution. However, ($-$)-epigallocatechin (EGC), ($-$)-gallocatechin gallate (GCG), and ($-$)-epicatechin (EC) bio-accessibilities were significantly decreased. A rat-fed study with an aqueous solution and TP nanoemulsion showed considerably low plasma concentrations of EGCG and

EGC. The data confirmed that a nanoemulsion for the tea polyphenols delivery might improve the absorption of EGCG through controlled release [40]. Heba Elmotasem and group aimed to innovate an effective oral sustained release of caffeine, a water-soluble drug. Due to its rapid absorption and elimination, caffeine is frequently used in administration to get an excellent therapeutic outcome. So, a *w/o* Pickering emulsion with caffeine incorporation was constituted from stabilized wheat germ oil using synthetic MgO nanoparticles (MgO NPS). Based on antioxidant hepatoprotective abilities, and anticarcinogenic, the components of the emulsion were selected. Such NPs of MgO were formed via the sol-gel process, and further characterization was done by TEM, cytotoxicity, contact angle, and X-ray diffractometry analysis. This Pickering emulsion stabilized by MgO NPs and conventional MgO particles was compared. Both methods evaluated caffeine release, stability, and droplet size. Earlier, a droplet size of $665.9 \pm 90$ nm was found stable in phase separation for 2 months. F1 could afford caffeine's sustained release following zero order kinetics that reached 70% within 48 h. About 36% hepatocellular carcinoma (HEPG2) growth inhibition has been shown by 100 ppm of F1. CCl4-intoxicated rats were used for in vivo and histopathological examinations. Liver enzymes (ALT and AST), inflammation marker (protein kinase C), and oxidative stress biomarkers targeted biochemical analysis suggested that the selected formula induces satisfactory hepatoprotection. The procedure has an economical approach in multiple therapies. It is safe, effective, and sustained levels of caffeine [41].

A study was carried out to examine the bacterial cellulose (BC) potential of melatonin (MLT) oral administration. It is a natural hormone and has issues such as poor solubility with low oral bioavailability. Bacterial cellulose got oxidized after sulfuric acid hydrolyzation to produce bacterial cellulose nanofiber suspension (BCNs). The emulsion solvent evaporation technique prepared melatonin-loaded BCNs (MLT-BCNs). These were characterized by XRD, FTIR, DSC, thermal analysis, SEM, fluorescence microscopy (FM), and instrument tools. The resulting data indicated that in BCNs, fibers became shorter and thinner than with BC. Both MLT-BCNs and BCNs have impressive thermodynamic stability; MLT was homogeneously distributed in MLT-BCNs. MLT-BCNs showed more speedy dissolution MLT rates compared with MLT in SGF and SIF, which are commercially available. The cumulative release rate of dissolution was found to be approximately 2.1 times that from MLT (commercially available), whereas in rats, it was 2.4 times more for the same. Hence, a promising delivery could be provided by MLT-BCNs with improved bioavailability and dissolution in the oral administration of MLT [42]. In vivo, cyclosporine ophthalmic emulsion (COE) with a spherical size distribution was studied; applied shear performance was affected by several physicochemical parameters such as zeta potential, pH, surface tension, and osmolality by the function of viscosity profile. A study was carried out using a modeling approach to predict drug bioavailability from COE to the conjunctiva, tear film breakup time, and cornea in human subjects as a function of the vehicle physicochemical properties such as surface tension, osmolality, and viscosity. From the bioavailability predictions, it was found that geometric mean ratios for test-to-reference in comparison to qualitatively and quantitatively formulations showed minor sensitivity. In contrast, the individual predictions were found to be sensitive to conjunctival permeability variations and corneal. The tear film breakup time from baseline values was found to be too sensitive to viscosity, showed slight susceptibility for surface tension, and was insensitive towards osmolality, as concluded from the parameter sensitivity analysis results. In addition, further enhancements in the modeling framework will develop the study to be more helpful in future prospects of COE bioequivalence in strong generic drug compounds [43].

Yamasaki et al. aimed to improve the oral bioavailability of praziquantel in conjugation with human serum albumin (HSA). These were prepared using praziquantel in an oil solution and HSA aqueous solution by spray drying. Amorphous praziquantel containing multiple smooth corrugated particles was aggregated and almost equivalent to the theoretical dosing. In an aqueous medium, the Praziquantel solubility was enhanced in the physical mixture and prepared particles. Moreover, the dissolution rate was also increased in the event of particles and not in physical combination. Hence, HAS addition increases the dissolution

rate when spraying through emulsification. The produced particles (HSA/praziquantel = 1/1 *w/w*) have higher concentration-time curve (AUC) values, of about two times, and maximum plasma concentration (Cmax) than raw praziquantel values, reported from a pharmacokinetic study. The particles' oral bioavailability was enhanced and was considered because of the increased dissolution rate. The process of praziquantel-HSA particle production could advance the oral bioavailability of other hydrophobic drugs [44]. The lipidic biocompatible and safe molecules are in high demand for self-micro-emulsifying drug delivery systems (SMEDDS). A work targeted to study oral mucosal irritation was reported to see the application of erucic acid-based bicephalous hydrolipid (BHL) in SMEDDS as an oil phase using Efavirenz (EFA), with poor bioavailability and water solubility of the drug. It showed higher drug loading efficiency, about $80.35 \pm 3.1\%$, with $0.23 \pm 0.031$ polydispersity index (PDI). EFA SMEDDS was also examined for standard stability tests and revealed that it was highly stable. EFA SMEDDS in vitro dissolution profile manifested > 95% release of drug in an hour and substantially enhanced bioavailability in vivo, at almost six times more than a drug in plain suspension. From the data outcomes, it was concluded that BHL could effectively be used as an oil phase in SMEDDS to improve the BCS Class II drug's solubility and bioavailability. In addition, it holds possibilities as a novel excipient for enhancements of solubility and bioavailability [45]. Candesartan cilexetil drug delivery faces a significant limitation of poor oral bioavailability, mainly due to its low solubility in aqueous solution and intestinal P-glycoprotein (P-GP) transporters effluxions. Yet, the P-gp extent role in decreased candesartan cilexetil oral bioavailability is cryptic. A study was carried out where previously developed candesartan cilexetil-loaded self-nano-emulsifying drug delivery system (SNEDDS) was examined for its ability to enhance oral bioavailability through intestinal P-GP transporters inhibition. P-GP–mediated efflux has some role in decreased candesartan cilexetil oral bioavailability despite whether or not the developed SNEDDS showed P-GP inhibition activity. Alternately, SNEDDS formulation with high surfactant concentration demonstrated a remarkable challenge in broad applications, specifically to chronically administered drugs. As the period of treatment increases, a reduction in intestinal mucosal damage was recorded from toxicity studies having acute and subacute designing. The surfactant-induced mucosal damage was found reversible from the observations. Hence, a sound delivery system with improved oral bioavailability could result from developed SNEDDS against chronically administered drugs [46].

### 4.9. Water Picric Emulsion Formulations

PE has received extensive attention for the encapsulation of lipophilic guests in the field of food and biomedicine. Although PE stabilities and demulsification control allow the release of species that have been encapsulated, they retain a challenge for the gastrointestinal tract. In a word, natural kaolinite with phosphatidylcholine was altered to prepare phosphatidylcholine-kaolinite to act as an emulsifier in stabilizing medium-chain triglyceride (MCT)/water PE that encapsulates curcumin. This was done to study curcumin-loaded-MCT-water PE for curcumin bioavailability and emulsification in a cell uptake assay and simultaneous implementation of intestinal digestion. The results suggested that phosphatidylcholine-kaolinite wettability would be modified by regulatory alteration temperature so the emulsion stability might be prevented. In the gastric acid presence condition, the prepared phosphatidylcholine-kaolinite has a contact angle of 123° of three-phase, which has optimal values for improved stabilization of the MCT/water PE. A condensed shell composition formed on the emulsion droplet surface from phosphatidylcholine-kaolinite, when it was dispersed in the W/O interface, controls demulsification efficiency from the release of encapsulated curcumin. After a period of 120 min of imitation gastric digestion, only 18.9% of the curcumin was released because of MCT/water PE demulsification. On the contrary, it was entirely removed after 150 min of simulated intestinal digestion, as expected. The PE stabilization by phosphatidylcholine-kaolinite is an encouraging transport carrier for drugs or lipophilic foods for improved bioavailability [47].

Na Man et al. aimed at the preparation of a myricitrin-loaded self-micro-emulsifying drug delivery system (MSMEDDS) for the enhancement of myricitrin's low oral bioavailability. MSMEDDS consisting of oil phase (ethyl oleate), (surfactant) Cremophor EL35, and, as a co-surfactant, dimethyl carbinol was prepared. The particle size, encapsulation efficiency, and zeta potential were used to characterize MSMEDDS. Prepared MSMEDDS exhibited a $21.68 \pm 0.15$ nm droplet with -ve zeta potential and high encapsulation efficiency of about $-23.17 \pm 1.03$ mV and 92.73%, respectively. M-SMEDDS are able to release myricitrin more significantly than free myricitrin, which was revealed from an in vitro release study. M-SMEDDS has a 2.47-fold increased relative oral bioavailability compared with free drugs. Both in vivo and in vitro studies demonstrated that M-SMEDDS could enhance the solubility of myricitrin and its oral bioavailability, providing preliminary confirmation for further M-SMEDDS [48] applications through clinical research. Darunavir-loaded lipid nanoemulsion was formulated in a research work to enhance its oral bioavailability and improve brain uptake. Darunavir was prepared from several lipid nanoemulsion batches by high-pressure homogenization using egg lecithin, Tween 80, and soya bean oil. DNE-3 was an optimized batch with a 109.5 nm globule size, $-41.1$ mV zeta potential, 93% entrapment efficiency, and 98% creaming volume. It was stable for 1 month at 4 °C with inconsiderable changes in spherical size and zeta potential ($p > 0.05$). Pharmacokinetics studies of in vivo male Wistar rats showed 223% Darunavir bioavailability compared with the suspension of the drug. DNE-3 has a two-fold higher Cmax and brain uptake than in suspensions found in an organ biodistribution study. Darunavir's increased bioavailability in nanoemulsions could lower the side effects related to the dose. Furthermore, because of high organ distribution, HIV reservoir organs have Darunavir passive uptake [49]. Amphiphilic bacterial cellulose nanocrystals (ABCNs) interfacial assembly by the PE method was suggested to enhance the compatibility with hydrophobic drugs and alginate. Biosynthesized bacterial cellulose was hydrolyzed by sulfuric acid to prepare BCNs used in particulate emulsifiers, considering alfacalcidol dissolved in $CH_2Cl_2$ as a model drug in the oil phase. The ultrasonic dispersion PE of O/W was formulated by and later well-dispersed in a solution containing alginate. By this, beads of drug-loaded alginate composite were prepared auspiciously by external gelation. BCNs have good colloidal properties, and a flocculated fibril network could be formed, beneficial for stabilizing Pickering emulsions revealed from results. BCNs have irreversible adsorption at the O/W interface, which could preserve PE droplets against Ostwald coalescence and ripening upon dispersing in an alginate-rich solution. Amphiphilic BCNs interfacial assembly and alginate composite beads hydrogel shells formed due to external gelation attain loading and sustained release of alfacalcidol. The release curves of alfacalcidol and the release mechanism from composite beads were significantly fitted in the Korsmeyer–Peppas model when associated with non-Fickian transport. Additionally, subsequent alginate composite beads can exhibit less cytotoxicity and advanced competencies for osteoblast differentiation [50].

Recent studies have revealed that to treat type 2 diabetes, metformin hydrochloride (Met) is a primitive drug and has the potential for Alzheimer's disease reduction. Met (B-Met-W/O/W SE) containing borneol W/O/W composite submicron emulsion was prepared with the expectation of better bioavailability, prolonged circulation time in vivo, and Met drug brain targeting. The optimized formulation has a mean droplet size of 386.5 nm, 0.219 polydispersity index, and 87.26% composite encapsulation potency. Met collaborated with carriers in B-Met-W/O/W SE, confirmed from FTIR analysis. Met in the B-Met-W/O/W SE in vitro release delivery system was slower than the Met-free drug. The AUC 1.27, MRT 2.49, and t1/2 of the B-Met-W/O/W SE system are 4.02-fold higher compared with the Met-free drugs, revealed from rat pharmacokinetic studies. B-Met-W/O/W SE system drug-targeting index to brain tissue was also more than that of the Met-W/O/W SE system and Met-free drug. The results concluded that the B-Met-W/O/W SE drug delivery system had encouraged candidature for clinical Alzheimer's disease treatment [51].

### 4.10. Nano System-Containing Formulations (NSCF)

NSCFs have recently shown significant advancement in nanotechnology research for active agents and drug delivery to human skin. A protein drug was extracted from tissues of medicinal leech and examined for kinetic stability, isotopic nanoemulsion formulation for topical delivery with negligible surfactant and co-surfactant amounts, and optimal stability and solubility were investigated in a study. Nanoemulsion formation and its stability were affected by oil phase physical properties. Several factors, such as oil content and type (sesame oil and olive oil), were evaluated for their impact on protein nano emulsion particle size and stability. In addition, protein nanoemulsion with optimized formulation was characterized for zeta potential, pH, viscosity, refractive index, droplet size, and transmission electron microscopy (TEM). For selecting the best formulation, stability studies were also carried out. The results concluded that an increase in sesame oil and olive oil concentration yielded nanoemulsions with some properties, such as higher stability and small-sized droplets. However, an olive oil-based nanoemulsion was observed to have slight alterations in droplet size. Several experiments selected a 25% olive oil-containing nanoemulsion as an optimized formulation as it has a tiny droplet size (143.1 nm), high zeta potential ($-33.3$ mV), and low polydispersity index. No significant changes were observed at 4 $^\circ$C for a 30-day storage duration in pH, viscosity, and droplet size. The technique also showed that the nanoemulsion selected was physically stable. Additionally, the particles have a spherical shape, morphologically confirmed from TEM studies. So, in conclusion, nanoemulsions of protein drugs have been proven as promising novel formulations and can improve protein drug stability significantly [52]. Sadaf Chaudhary et al. demonstrated that a self-nano-emulsifying drug delivery system (SNEDDS) improves Nabumetone (NBT) oral bioavailability and anti-inflammatory impact. NBT has poor solubility in aqueous drugs exploring less bioavailability when orally administrated. The preparation of NBT-SNEDDS was done using a pseudo-ternary phase diagram and polyethylene glycol-400 (PEG-400), Capryol-90, and Tween-80. SNEDDS components were curtained, and for deionized water, PEG-400, Tween-80, and Capryol-90 in an optimal ratio of 58:4:16:22 was found to be optimal. The prepared SNEDDS underwent characterization for anti-inflammatory and pharmacokinetics properties and compared with optimized NBT-SNEDDS and the marketed tablet suspension in rats. The SNEDDS drug had a 3.02-times higher oral bioavailability than the marketed tablet suspension. A significant increase in anti-inflammatory activity was presented by NBT-SNEDDS than commercial NBT products that were orally administered. NBT-SNEDDS could be a potent carrier for NBT oral dosing with improved bioavailability and therapeutic impacts suggested from the results [53]. Resveratrol (RVT) re-dispersible dry emulsion (DE) was prepared using caprylic/capric glyceride (CCG) and using low-methoxy pectin (LMP) as the lipid phase comprising component and emulsifier. A Box–Behnken design was used to optimize and examine redispersed emulsion size, spraying efficiency, and angle of repose from spray dryer pump speed and formulation effects. For the estimation of RVT-DE dissolution properties, redispersibility was used. LMP and CCG concentrations affect redispersed emulsion size explored from the results. Any change in concentration of LMP and CCG, i.e., increase or reduction, results in a small droplet size in the emulsion and rapid drug dissolution from RVT-DE and influences the repose angle. High CCG and low LMP concertation using RVT-DE generation explored less repose angle, suggesting suitable flow property. The prepared formulation was optimized within the design space with 7% *w/w* of CCG and 2.75% *w/w* of LMP when sprayed at a pump speed of 10.1 mL/min, significantly fulfilling all criteria, i.e., high spraying efficiency, good flow, and small redispersed size. From intact RVT, RVT in RVT-DE has extraordinarily high photostability [54].

Organogels used in pharmaceutical and food sciences have some technical issues, such as confined drug diffusion and insufficient proper gelating molecules. They are necessary features to design new products. The use of emulsions is an alternative for enhancing the technological properties of organogels. There is a need for more information about the permeability and bio-accessibility behavior of bioactive-loaded, organogel-based emulsions. To study the physical properties of betulin, curcumin, and quercetin, three different

bioactive-loaded vegetable oil-containing organogel-based emulsions, experimental work was carried out for bio-accessibility and influence. Coconut oil, canola, and myverol were used as a gelator (10% *w/w*) to prepare organogels by mixing water (80 °C) and melting proper organogels at high shear conditions (20,000 rpm) Water-in-oil emulsions (at 5, 10 and 12.5 wt.% of water content) were formulated. Rheological tests (frequency, creep-compliance measurements, temperature sweeps, and amplitude), micrographs, particle size, and DSC analysis were performed. Lipolysis, bio-accessibility, in vitro digestion, and permeability assays on the Caco-2 cell culture were also investigated. Coconut oil-based organogels have poor emulsification properties [55]. A multiple W/O/W emulsion (ME) was formulated from clotrimazole (CLT) and examined for anticandidal agent efficacy against the marketed products. Physicochemical characterization was carried out from the estimated CLT-ME selected previously. Franz diffusion cells were used with human skin, sublingual and vaginal mucosae, and porcine buccal biological membranes to assess in vitro liberation and ex vivo permeation behavior. Antifungal activity was also tested against *Candida* strains. CLT-MEs with two different sizes, 29.206 and 47.678 μm, exhibited high zeta potential of −55.13 and −55.59 mV, with skin-compatible pH values of 6.47 and 6.42 and dependency on pH variation. CLT-MEs showed physicochemical stability at room temperature and were kept up to 180 days. CLT-MEs have exhibited pseudoplastic behavior with viscosities and hysteresis areas of 331 mPa·s and 286, with high spreadability properties to commercial products. An enhanced CLT release pattern was contributed with the ME system with a hyperbolic model following. Compared with commercial products, CLT with high skin permeation flux was from the ME system. Compared with commercial reference, more CLT amounts were retained in mucosae and skin. CLT-MEs have high antimycotic efficacy, so they could become an excellent tool for topical candidiasis treatments and clinical investigations [56].

José Soriano-Ruiz et al. conducted a study to concur bisdemethoxycurcumin (BDMC) low bioavailability and solubility by preparing a self-micro-emulsifying system loaded with BDMC (BDMC-SMEDDS). Pseudo-ternary phase diagrams (PTPDs), compatibility and solubility tests, and d-optimal concepts were used for formulation designing. In vitro assessment of fabricated BDMC-SMEDDS was done to determine entrapment efficiency (EE), droplet size (DS), morphology, drug stability, and release. Moreover, in vivo behavior examination was also done in rats after BDMC-SMEDDS oral administration. The resultant formulation contained BDMC (50 mg), ethyl oleate (EO, oil, 207.5 mg), Kolliphor EL (K-EL, as an emulsifier, 645.3 mg), and PEG 400 (co-emulsifier, 147.2 mg). Good stability of BDMC-SMEDDS was found, with a mean size of $21.25 \pm 3.23$ nm and $98.31 \pm 0.32\%$ EE. The optimal formulation was found to compose of Kolliphor EL (K-EL, emulsifier, 645.3 mg), PEG 400 (co-emulsifier, 147.2 mg), ethyl oleate (EO, oil, 207.5 mg), and BDMC (50 mg). The BDMC-SMEDDS with good stability had a mean size of $21.25 \pm 3.23$ nm and EE of $98.31 \pm 0.32\%$. From BDMC-SMEDDS, around 70% of BDMC was released within 84 h than free BDMC, at <20%. In particular, the in vivo behavior of BDMC-SMEDDS showed that BDMC plasma concentration and AUC (0–12 h) were increased when compared with free BDMC. In all respects, BDMC-SMEDDS is potent in improving BDMC bioavailability and solubility and could be applicable in clinics [57]. Another study targeted luteolin's oral bioavailability and solubility improvement through supersaturated self-nano-emulsifying drug delivery system (S-SNEDDS) employment. SNEDDS formulation is composed of polyethylene glycol 400, caprylic/capric triglyceride, and castor oil hydrogenated with polyoxyl 35 with a ratio of 31.7:20.1:48.2 by weight. It was optimized and determined from pseudo-ternary phase diagrams, solubility studies, and central composite design. For luteolin-loaded SNEDDS at a 2% mass ratio, hydroxypropyl methylcellulose (HPMC) K4M was found to be an optimal precipitation inhibitor based on in vitro precipitation evaluations. A nanoemulsion having a 25.60 nm particle size was formed from luteolin S-SNEDDS and has a −10.2 mV zeta potential after dilution. Luteolin and HPMC K4M interactions were examined by differential scanning calorimetry (DSC), (FTIR) Fourier-transform infrared spectroscopy, powder X-ray diffraction (XRD), and 1H NMR spectroscopy.

Additionally, an outstanding 99% in vitro dissolution has been achieved by S-SNEDDS at pH 6.8 in phosphate buffer with 0.5% Tween 80. A S-SNEDDS pharmacokinetics study in vivo revealed a more noteworthy luteolin oral bioavailability enhancement (2.2-fold) in rats than in conventional SNEDDS. In conclusion, from the data illustrated, S-SNEDDS technology could be applicable at most minuscule for luteolin in promoting oral bioavailability and solubility for poorly water-soluble drugs [58]. For the treatment of osteoporosis, a novel cathepsin K inhibitor, HL235, has been designed and synthesized. To improve HL235 oral bioavailability, SMEDDS was designed to overcome HL235's low aqueous solubility. For the selection of a suitable oil, surfactant, and cosurfactant, a HL235 solubility study was conducted. Pseudo-ternary phase diagrams were prepared to identify the components' range in the isotropic environment and microemulsion region. To optimize the formulation of SMEDDS, desirability function and D-optimal mixture design were interrogated to get requisite physicochemical features, such as high solubilization capability and higher drug concentration after 15 min of dilution with simulated gastric fluid (SGF); this led to the formulation and optimization of HL235-loaded SMEDDS composed of surfactant (75.0% Tween 20), cosurfactant (20.0% Carbitol), and oil 5.0% Capmul MCM EP. The microemulsion formulated has been optimized, and the droplet size was $10.7 \pm 1.6$ nm with a spherical shape. Furthermore, there was a 3.22-fold elevated SMEDDS formulation relative oral bioavailability in rat pharmacokinetic studies than in its DMSO: PEG400 (8:92, *v/v*) solution. A promising approach could be made available by SMEDDS formulations on optimization by D-optimal mixture design to improve HL235 oral bioavailability [59].

Table 1 summarizes the different types of emulsion formulations and their impacts.

**Table 1.** Different types of emulsion formulation and their uses.

|  | Emulsion Type | Uses | Ref |
|---|---|---|---|
| 1 | Silybin nanocrystal self-stabilized Pickering emulsion, core-shell arrangement with 27.3 µm droplet, 40 days stability. | Increase oral bioavailability of silybin | [16] |
| 2 | (Poly lactide-co-glycolide) microparticles standard Emulsion | For a controlled drug delivery system | [17] |
| 3 | Silymarin-loaded nanoemulsions with Capryol 90 | Improve oral bioavailability | [18] |
| 4 | Phosphorylated tocopherols-based emulsion | make poorly water-soluble drugs soluble | [19] |
| 5 | Polyethylene glycol, polysorbate, and capric triacylglycerol/caprylic-based emulsion | For delivering ellagic acid | [21] |
| 6 | Protein formulation in lyophilized electrospun fibers | Increases therapeutic compounds' shelf life | [22] |
| 7 | Capsule designing with insulin-like hydrophilic drugs | Preservation of biological activity and stability | [23] |
| 8 | Mixed phosphatidylcholine micelles encapsulation | Increase the bioavailability of sesamolin | [24] |
| 9 | Betulinic acid-loaded nanoemulsions with labrasol | Increase hepatoprotective activity and bioavailability of betulinic acid | [27] |
| 10 | Emulsion polymer matrices | Polymer degradability | [29] |
| 11 | Nanoemulsion from surfactant-free Pickering formulations | Enhancing oral bioavailability | [31] |
| 12 | Emulsion with collapse protectant (glycine) as an aqueous phase, 4% alginate/gelatin containing mannitol and synperonic PE/P 84 (surfactant) as an oil phase | Lyophilized disintegrating tablet | [32] |
| 13 | Smart-releasing emulsion (*o/w*) | Controlled drug delivery | [33] |
| 14 | Solid-self-nano-emulsifying-drug-delivery-systems | Enhancing the simvastatin bioavailability and dissolution rate | [34] |
| 15 | Self-nano emulsifying system | Improve the bioavailability of boswellic acids | [36] |
| 16 | Emulsion-solvent-freeze-drying/evaporation | Stable Ibuprofen nanoparticles formulation | [38] |
| 17 | W/O/W emulsion and layer-by-layer electrostatic adsorption | Formation of PLGA/Bengal colloidal particles | [39] |
| 18 | Produce bacterial cellulose nanofiber suspension | Enhancing oral bioavailability of bacterial cellulose | [42] |
| 19 | Emulsion based on praziquantel in an oil solution and HSA aqueous solution by spray drying | Improve the oral bioavailability of praziquantel | [44] |

**Table 1.** *Cont.*

|  | Emulsion Type | Uses | Ref |
|---|---|---|---|
| 20 | Self-micro-emulsifying drug delivery systems | Improving the bioavailability and water solubility of the drug | [45] |
| 21 | Phosphatidylcholine-kaolinite acts as an emulsifier in stabilizing medium chain triglyceride (MCT)/water | Improved bioavailability of curcumin | [47] |
| 22 | Myricitrin-loaded self-micro-emulsifying drug delivery system | Myricitrin low oral bioavailability enhancement | [48] |
| 23 | Emulsion based on Amphiphilic bacterial cellulose nanocrystals | Enhance the compatibility with hydrophobic drugs and alginate | [50] |
| 24 | Self-nano-emulsifying drug delivery system | Improves Nabumetone (NBT) oral bioavailability and anti-inflammatory | [53] |
| 25 | W/O/W emulsion from clotrimazole | Anticandidal agent efficacy against marketed products | [56] |
| 26 | Self-micro-emulsifying system | Improving bioavailability and solubility of bisdemethoxycurcumin | [57] |
| 27 | Supersaturate self-nano-emulsifying drug delivery system | Luteolin's oral bioavailability and solubility improvement | [58] |

## 5. Conclusions

The bioavailability and solubility of hydrophobic and lipophilic drugs is an issue for their oral administration, which is resolved using self-emulsifying formulations to a certain extent. Thus, this review article deals with emulsion-formulation-based drug delivery and improving the bioavailability of hydrophobic and lipophilic drugs. The dispersion or loading efficacy of the drug into emulsions depends upon the emulsion's constituents, such as oil, surfactants, and co-surfactants. In this regard, the nanoemulsions have been found effective for target drug delivery and improving the bioavailability of poorly water-soluble drugs. The present scenario of progression in technologies for drug carrying has broadcast the expansion of innovative drug carriers for the target release of self-emulsifying pellets, tablets, capsules, and microspheres, which has improved drug delivery with self-emulsification. Thus, the present review article will help readers working in the field of emulsion-based drug delivery with the increased bioavailability of lipophilic/hydrophobic drugs at the current time.

**Author Contributions:** R.K.A. wrote manuscript and prepared concept, K.S. checked grammar and plagiarism, A.B. proposed concept and executed the idea of this review article. All authors have read and agreed to the published version of the manuscript.

**Funding:** This research received no external funding.

**Acknowledgments:** Authors are greatly thankful to Kadi Sarva Vishwavidhyalaya, Gandhinagar, India, for support and infrastructure facilities. A.B. acknowledges TWAS-UNESCO Associateship-Ref. 3240321550 for providing opportunities to visit the Department of Chemistry, Indian Institute of Technology Madras, Chennai, India.

**Conflicts of Interest:** The authors declare no conflict of interest.

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
