# Peer review of "Recent Advances in Improving the Bioavailability of Hydrophobic/Lipophilic Drugs and Their Delivery via Self-Emulsifying Formulations"

_colloids, doi:10.3390/colloids7010016_

Round 1

Reviewer 1 Report

The article provides an overview of the field of SEFs. This field is very exciting and always worth a new overview. However, from my point of view, the paper does not succeed in this. 

I find the introduction very weakly formulated. Overall, it is difficult to find a common thread in the text and individual passages are also difficult to understand. A general introduction to the topic is missing, but specific topics such as polymers are taken up very quickly.

Why are some words capitalized or in a different font?

Abbreviations are used without introducing them (e.g. EFDDS).

Tables or graphs would enhance the paper significantly.

Single bullet point feedback on individual parts.

L13: What is surfactant dissipation?

L53: A literature review is mentioned without giving the literature.

Paragraph 3.3 is well worded.

Paragraph 4 is way too long and needs substructures.

L 487: Where is the EE described?

The summary reflects the problem of the paper. It does not contain any added value because the red thread and thus essential statements are missing.

Author Response

Reviewer 1:

The article provides an overview of the field of SEFs. This field is very exciting and always worth a new overview. However, from my point of view, the paper does not succeed in this. I find the introduction very weakly formulated. Overall, it is difficult to find a common thread in the text and individual passages are also difficult to understand. A general introduction to the topic is missing, but specific topics such as polymers are taken up very quickly.

  • Now, the introduction is made easy to understand.

Why are some words capitalized or in a different font?

  • Now, such changes are made.

Abbreviations are used without introducing them (e.g. EFDDS).

  • Now, it is done.

Tables or graphs would enhance the paper significantly.

  • We have included table as well as graph in the revised manuscript.
  • Single bullet point feedback on individual parts.
  • Now these are mentioned in the revised manuscript.

L13: What is surfactant dissipation?

  • Surfactant dissipation is related to surfactant solubility.

L53: A literature review is mentioned without giving the literature.

  • Now it is provided.

Paragraph 3.3 is well worded.

  • Thank you for this appreciation.

Paragraph 4 is way too long and needs substructures.

  • Now it is reduced.

L 487: Where is the EE described?

  • Now it is described.

The summary reflects the problem of the paper. It does not contain any added value because the red thread and thus essential statements are missing.

  • In conclusion section, now it is mentioned.

Reviewer 2 Report

The reviewer assessed the manuscript entitled "Recent advances in improving the bioavailability of hydrophobic/ lipophilic drugs and their delivery via self-emulsifying formulations".

This reviewer considers this review well-organized and structured and collects recent investigations in the field. Therefore, I think that this review fulfills the standards of the Colloids and Interfaces journal, but the authors need to consider some minor corrections before the publication of the manuscript:

-  In the title, the review considers that it should be “hydrophobic/lipophilic” instead “hydrophobic/ lipophilic”.

-   In the abstract and the introduction, the text formatting must be unified.

-   The authors should correct typos such as the one in line 263.

-   Citations should be made according to the journal format. Please, review: B. Mutaliyeva et al. (line 305); Ranjit K. Harwansh and coworkers (line 338) and Keishi Yamasaki et al. (line 578).

-   Please, correct “Mg (OH)2” by “Mg(OH)2”.

Author Response

Reviewer 2:

The reviewer assessed the manuscript entitled "Recent advances in improving the bioavailability of hydrophobic/ lipophilic drugs and their delivery via self-emulsifying formulations".

This reviewer considers this review well-organized and structured and collects recent investigations in the field. Therefore, I think that this review fulfills the standards of the Colloids and Interfaces journal, but the authors need to consider some minor corrections before the publication of the manuscript:

  • Thank you very much for appreciation us.

In the title, the review considers that it should be “hydrophobic/lipophilic” instead “hydrophobic/ lipophilic”.

  • Now it is done.

In the abstract and the introduction, the text formatting must be unified.

  • Now it is done.

The authors should correct typos such as the one in line 263.

  • Now it is done.

Citations should be made according to the journal format. Please, review: B. Mutaliyeva et al. (line 305); Ranjit K. Harwansh and coworkers (line 338) and Keishi Yamasaki et al. (line 578).

  • Now it is done.

Please, correct “Mg (OH)2” by “Mg(OH)2”.

  • Now it is done.

Reviewer 3 Report

The manuscript under review reports recent advances in self-emulsifying formulations for improving drug bioavailability. To help the reader I would suggest reorganizing Chapter 4 "Recent Advances" into paragraphs because the Chapter is too long. In addition, I really encourage authors to summarize Chapter 4 in Tables. In addition, the mechanism for self-emulsification could be efficiently described by a figure.

Author Response

Reviewer 3:

The manuscript under review reports recent advances in self-emulsifying formulations for improving drug bioavailability. To help the reader I would suggest reorganizing Chapter 4 "Recent Advances" into paragraphs because the Chapter is too long. In addition, I really encourage authors to summarize Chapter 4 in Tables. In addition, the mechanism for self-emulsification could be efficiently described by a figure.

  • Now chapter 4 has been reorganized into paragraphs. In this revised manuscript, now we summarized findings in the form of table. We tried to describe the mechanism for self-emulsification by a graph.